# Preparation of Gelatin from Broiler Chicken Stomach Collagen

**DOI:** 10.3390/foods12010127

**Published:** 2022-12-27

**Authors:** Aneta Prokopová, Robert Gál, Pavel Mokrejš, Jana Pavlačková

**Affiliations:** 1Department of Polymer Engineering, Faculty of Technology, Tomas Bata University in Zlín, Vavrečkova 275, 760 01 Zlín, Czech Republic; 2Department of Food Technology, Faculty of Technology, Tomas Bata University in Zlín, Vavrečkova 275, 760 01 Zlín, Czech Republic; 3Department of Lipids, Detergents and Cosmetics Technology, Faculty of Technology, Tomas Bata University in Zlín, Vavrečkova 275, 760 01 Zlín, Czech Republic

**Keywords:** biotechnology, chicken stomachs, collagen, enzyme conditioning, food, gelatin, meat by-products, pharmacy, proteins

## Abstract

With the increasing consumption of poultry meat around the world, the use of chicken stomachs as a source of collagen is being offered. The objective of this study was to extract gelatin from the stomachs of broiler chickens and to estimate their gel strength, ash content, viscosity, gelling point, melting point, clarity and digestibility. An innovative biotechnological method based on the conditioning of collagen with a microbial endoproteinase (Protamex^®^) and hot-water extraction was used to control the chemical and thermal denaturation process of collagen to prepare gelatin. The experiments were planned using a Taguchi design, 2 factors at 3 levels; factor A for the amount of proteolytic enzyme (0.10, 0.15 and 0.20%) and factor B for the extraction temperature (55.0, 62.5 and 70.0 °C). Data were statistically processed and analyzed at a significance level of 95%. The gelatin yield averaged 65 ± 8%; the gel strength ranged from 25 ± 1 to 439 ± 6 Bloom, the viscosity from 1.0 ± 0.4 to 3.40 ± 0.03 mPa·s, gelling point from 14.0 ± 2.0 to 22.0 ± 2.0 °C, melting point from 28.0 ± 1.0 to 37.0 ± 1.0 °C. The digestibility of gelatin was 100.0% in all samples; the ash content was very low (0.44 ± 0.02–0.81 ± 0.02%). The optimal conditions for the enzymatic treatment of collagen from chicken stomachs were achieved at a higher temperature (70.0 °C) and a lower amount of enzyme (0.10–0.15%). Conditioning chicken collagen with a microbial endoproteinase is an economically and environmentally friendly processing method, an alternative to the usual acid- or alkaline-based treatment that is used industrially. The extracted products can be used for food and pharmaceutical applications.

## 1. Introduction

Collagen is a protein biomolecule made of amino acids. Collagen is the most abundant structural protein in the extracellular matrix of various connective tissues in the body, responsible for strength and flexibility [1,2]. Hydrolysis is used for the biochemical breakdown of collagen bonds. In an acidic environment, it is assumed that an electrophilic mechanism of hydrolysis occurs [3] in comparison to the nucleophilic mechanism that is common in an alkaline environment [4]. The extraction of gelatin based on the partial acid-controlled hydrolysis of the collagen structure is called type A gelatin, and the second one based on partial alkaline-controlled hydrolysis of the collagen structure is called type B gelatin. Both chemical methods are slow and costly and have an environmental footprint [5]. Compared to chemical agents, enzymes are more environmentally friendly, minimize production costs, and allow the desired functional properties of gelatins to be achieved [6]. The most used enzymes today include industrially produced microbial enzymes, animal enzymes trypsin and pepsin, as well as plant enzymes (e.g., papain) [7]. Enzymes are pure proteins, are fully biodegradable, and do not produce unwanted by-products [8].

The production of gelatin consists of two main technological steps: the raw material pretreatment (conditioning) and the heat extraction process. Structurally, there is a transition from a complex spiral to a random coil. The final transformation occurs during the thermal extraction process as a result of the splitting of hydrogen and covalent bonds [9]. Individual chains of tropocollagen molecules are released into the aqueous solution, and this creates a solution of collagen in water (gelatin) [10]. At lower temperatures, some of the bonds between the chains are restored, and sol–gel transition occurs (a gelatin gel is formed). The yield of gelatin depends on the collagen source and the number of cross-linked covalent bonds. Gelatin properties are influenced by the species, gender, and age of the initial tissue and process conditions (temperature, time, pH of the environment) [11].

The microbiological safety, appearance, smell, color, taste, physicochemical, rheological, and functional properties of gelatin depend primarily on the source and type of collagen [12]. Gelatins have the ability to bind large amounts of water and form thermo-reversible gels with melting points close to the human body temperature (approximately 35–39 °C) [13]. The most important parameter for determining the quality of gelatin is the strength of the gelatin gel, which is influenced by hydrogen bonds between water molecules and free carboxyl groups of amino acids. Gelatins are used in the food, pharmaceutical, medical, and also cosmetic industries [14,15]. The sources of commercial gelatin are mainly pork and beef skin and beef bones [16], and the rest are alternative sources (e.g., poultry skin, bones, or fish skin) [17,18,19,20,21,22]. Compared to fish gelatin, poultry gelatin does not have an unpleasant smell [23].

The consumption of poultry meat continues to grow, increasing the production of by-products with a high percentage of protein [24]. Chicken stomachs represent up to 3% of poultry live weight [25,26]. According to statistics, 677,200 tons of gelatin were produced from pork and beef in 2021. In the following years, it is assumed that the subsequent annual consumption of gelatins will be 8% higher than the previous one [27]. There is an estimate that, around 2035, the production of beef and pork gelatin will no longer be sufficient to cover the world market. Therefore, the production of gelatin from alternative collagen tissues [28] will be desirable. In general, the valorization of animal by-products from slaughterhouses and the retail sector will be a high priority in the management of solid waste [29,30]. Poultry stomachs are considered waste in some countries and are therefore treated as such. At the same time, it appears to be a very promising tissue that could be used to extract gelatin and compete with commercial products [31,32]. Our team deals with the enzymatic processing of poultry tissues (e.g., feet, bones, and heads) into gelatins [23,33,34].

The objective of the work is to verify the process conditions for the preparation of gelatins from collagen from poultry stomachs (conditioned with enzyme protease) and to monitor the effect of selected process parameters on the yield of gelatins. Subsequently, the gelatin gel strength, ash content, viscosity, melting, and gelling points are measured, as is its digestibility. Furthermore, we propose optimal conditions for the processing of poultry stomachs into gelatins. Hypothesis: Under the chosen process conditions of multistage gelatin extraction, it is possible to prepare gelatins from poultry stomachs with properties suitable for use in the food industry.

## 2. Materials and Methods

### 2.1. Laboratory Equipment

Electronic analytical laboratory balance Kern 770 (Kern GmbH, Bensheim, Germany), dryer Memmet ULP 400 (Memmert GmbH + Co. KG, Büchenbach, Germany), heating nest LTHS 250 a 500 (Merci, Brno, Czech Republic), meat cutter SPAR Mixer SP–100 AD–B (Gastrotip, Hradec Králové, Czech Republic), heating plate Schott Garate GMBH with a magnetic stirrer (Schott, Berlin, Germany), dryer WTB Binder E-28-TB1 (Binder, Baddechenstedt, Germany), shaker LT3 Kavalier (Kavalier, Praha, Czech Republic), pH meter WTW 526 (WTW, Weilhein, Germany), muffle furnace Nabertherm L9/11 (Nabertherm, Lilienthal, Germany), hob with thermostat and magnetic stirrer IKA C MAG HS7 (IKA-Werke, Staufen im Breisgau, Germany), Sevens-LFRA gel strength analyzer (Leonard Farnell and Co., Ltd., Hampton, UK), vertical mixer ETA 0010 Nová Linie (ETA, Praha, Czech Republic), magnetic stirrer IKA Labortechnik PCT Basic (IKA-Werke, Staufen im Breisgau, Germany), KRUPS mill (KRUPS, Praha, Czech Republic), blender BOSH (BOSH, Gerlinger Schillerhöhe, Germany), refrigerator Samsung (Samsung, Praha, Czech Republic), desiccator (Merci, Brno, Czech Republic), Ubbelohde viscometer (Verkon, Praha, Czech Republic), self-sealing LDPE bags (McPen, Děčín, Czech Republic), test tubes (Merci, Brno, Czech Republic), F57 digestibility bags (Dräger Medical, Lübeck, Germany), incubation bottle (Dräger Medical, Lübeck, Germany).

Distilled water, acetone, 0.2 mol/L NaCl, 0.06 mol/L NaOH, ethanol and petroleum ether in a ratio of 1:1, chloroform (Faren, Uherské Hradiště, Czech Republic), pepsine (P-LAB, Praha, Czech Pepublic), pancreatin (P-LAB, Praha, Czech Pepublic). Protamex^®^ (Novozymes, Bagsvaerd, Denmark): a microbial proteinase developed for the hydrolysis of food proteins, declared activity: 1.5 AU/g, optimal processing conditions: pH 5.5 to 7.5, temperature 35.0 to 60.0 °C, enzyme inactivation at 85.0 °C for 10 min. Protamex^®^ is an endopeptidase that cleaves the bonds inside the protein structure, allowing higher molecular weights of the peptide chains [35].

### 2.2. Chicken Stomachs

Ross 708 broiler chicken stomachs, aged 35 days, were provided by a regional company (Raciola Ltd., Uherský Brod, Czech Republic). Stomachs were cleaned, rinsed, ground into 3 mm particles, and homogenized in an industrial meat cutter (meat cutter SPAR Mixer SP–100 AD–B, Gastrotip, Hradec Králové, Czech Republic). Subsequently, the stomachs were deep-frozen and kept in a freezer at −20.0 ± 1.0 °C; before the experiments, the material was thawed in a refrigerator overnight at 5.0 ± 1.0 °C. After the mixture thawed, excess water and blood were filtered. The composition of the initial tissue was the following: dry matter content 19.10 ± 0.05%; in dry matter: protein 75.6 ± 0.8%, fat 21.70 ± 0.01% and ash 3.900 ± 0.005%.

### 2.3. Processing of Chicken Stomachs into Gelatins

Figure 1 shows a flow chart of the key steps in the processing of collagen from chicken stomachs into gelatins.

The preparation of purified collagen from chicken stomachs was performed in the following steps:Washing of the stomachs with running cold water (approximately 10 min) to remove albumins and impurities from the tissue.Processing stomachs in a 0.2 mol/L NaCl solution in a 1:6 ratio at laboratory temperature for 1.5 h to remove globulins from the structure.Processing of stomachs in a 0.06 mol/L NaOH solution in a ratio of 1:6 at laboratory temperature for 8 h; then the NaOH solution is replaced with a fresh one and the same treatment is carried out for 16 h. At this point, the noncollagenous substances are removed.Gentle drying of the stomachs at 35.0 ± 1.0 °C for 36 h.Degreasing of stomachs with petroleum ether/ethanol solvent mixtures (mixed in a 1:1 ratio) in a 1:9 ratio (tissue: solvent mixture) for 36 h. During this time, the solvent is changed twice for a fresh one.Grinding of desiccated purified collagen in smaller batches to 1 mm particles.

The preparation of gelatins from purified collagen was performed in the following steps:Collagen conditioning with Protamex^®^ enzyme; collagen and distilled water mixed in a 1:10 ratio. The solution is gently shaken for approximately 20 min and then the pH is adjusted to 6.5 ± 0.5 using dilute solutions of acids or alkalis.The addition of proteolytic enzyme to the mixture in the amount according to factor A (the amount is related to the dry matter of the purified collagen, 90.2%).Shaking the mixture at room temperature (22.0 ± 1.5 °C) for 24 h; during the first 4 h, the pH value is checked and adjusted.Filtration through a sieve equipped with PA fabric; collagen hydrolysate is obtained. The hydrolysate is heated to 85.0 ± 1.0 °C (enzyme inactivation) and dried in a thin layer (approximately 4 mm) at 60.0 ± 1.0 °C for 48 h.Washing the conditioned collagen (approximately 10 min) under cold running water to remove the present enzyme.The extraction of the first fraction of gelatin in distilled water (in a collagen: water ratio = 1:8) at a temperature according to factor B for 45 min.After filtering a gelatin solution of the first fraction, the remaining collagen is subjected to the second extraction of gelatin in distilled water (in a 1:7 ratio) at 80.0 ± 1.0 °C for 45 min.Third extraction of gelatin in distilled water (in a ratio of 1:7) at 90.0 ± 1.0 °C for 45 min.Gelatin solutions of the first and second fractions heated to 85.0 ± 1.0 °C and maintained for 10 min to inactivate any remaining enzyme.Gelatin solutions poured onto a metal sheet and dried in a thin film (approx. 4 mm) at 53.0 ± 1.0 °C for 24 h.The undissolved solid residue is dried at 103.0 ± 2.0 °C for 24 h.

### 2.4. Experimental Design

The experiments were carried out according to the Taguchi design with two factors at three levels. Taguchi design is a multifactorial experiment method, which allows the examination of the minimum number of measurements needed for statistical evaluation and, at the same time, minimizes the cost of the process conditions [36,37]. A total of 10 experiments were carried out, of which there were 9 experiments and one blind experiment without enzyme addition to estimate the effectiveness of enzymatic processing. Factor A represented the amount of enzyme added (0.10, 0.15 and 0.20%) and factor B represented the extraction temperature (55.0, 62.5 and 70.0 °C). The factor values are based on our previous study, in which the amount of enzyme added and the extraction temperature were found to influence gelatin extraction and its quality [23].

The results of the analyses that were carried out according to the standard testing methods for edible gelatin were processed in the Microsoft Office Excel program (Denver, CO, USA, 2010) and then evaluated according to the statistical program Minitab^®^ 19 (Fujitsu Ltd., Tokyo, Japan). All gelatins from the first fractions were analyzed for gelatin gel strength, ash content, melting point, gelling point, clarity, viscosity, and digestibility. Due to the small yields of gelatins of the second and third fractions, all samples of the second and third fractions were mixed (each separately). The same analyses as for the gelatins of the first fractions were performed on these 2 mixtures. According to the p-factor, the significance level was determined with 95% probability.

### 2.5. Calculation of the Results

Table 1 summarizes the list of equations used for calculations in the experiment.

#### 2.5.1. Dry Matter Content

The dry matter content of the gelatins was determined after placing the sample in a drying oven with air circulation at 103.0 ± 2.0 °C and drying to a constant weight. The dry matter content, *DM* (%), was calculated according to Equation (1), *m*_1_ is the weight of the sample after drying (g) and *m*_2_ is the weight of the sample before drying (g) [38].

#### 2.5.2. The Yield of Gelatins

The yield of gelatins (the extraction efficiency) was estimated according to Equation (2), where *η* is the yield of gelatins (%), *L* is the dry matter of the extract (g), and *DW* is the dry weight of the initial issue (g). For all of the yields, the balance error of the measurement BE (%) according to Equation (4) was calculated. The starting point for its calculation is Equation (3), *TOTAL BALANCE* (%), where *INTUP* is collagen dry matter (g) and *EXIT* is the sum of the yields of all gelatin fractions, including the hydrolysate and the undissolved solid residue (g).

#### 2.5.3. Ash

The ash content was determined as follows. The sample was annealed in a muffle furnace at 650.0 ± 5.0 °C for 4 h. After cooling, the ash was weighed, and the ash content of the samples was calculated according to Equation (5), where *AS* is the ash content (%), *m_A_* is the weight of the ash (g), and *m* is the weight of the sample (g) [38].

#### 2.5.4. Gel Strength

The gel strength of gelatin measures the rigidity of a gel formed from a 6.67% solution prepared according to the prescribed conditions; 7.5 g of gelatin was mixed with 105.0 mL of distilled water in a standardized Bloom jar. The gelatin was first swollen for 20 min and then dissolved at 60.0 ± 1.0 °C in a water bath for approximately 10 min. The gelatin solution was cooled to laboratory temperature and placed in an incubator for 24 h at 10.0 ± 1.0 °C. The Bloom value was measured as the force (weight in grams) required to depress the prescribed area of the sample to a depth of 4 mm using the Stevens LFRA texture analyzer (Leonard Farnell and Co., Ltd., Hampton, UK) [38].

#### 2.5.5. Viscosity

The determination of the viscosity of the 6.67% gelatin solution was estimated by measuring the flow time with a standardized pipette at 60.0 ± 1.0 °C. Measurement was carried out on a Ubbelohde viscometer, and measured flow time was converted to viscosity by substituting in Equation (6), where *ν* is viscosity (mPa·s); *c* is viscometer constant estimated by verified calibration fluid (0.5); *B* is correlation constant to the kinetic energy estimated from the dimensions of the viscometer (2.8); *t* is the arithmetic average of the measured flow times (s), and ρ is the density of the gelatin solution (1.005 g/cm^3^) [38].

#### 2.5.6. Clarity

The clarity of a 6.67% gelatin solution was determined at 45 °C by measuring the percentage transmittance through a 1 cm cuvette at 640 nm. Before measurement, calibration was performed with distilled water [38].

#### 2.5.7. Melting Point

The method according to Moosavi-Nasab [39] with some modifications was used to estimate the melting point. A gelatin solution at the same concentration (6.67%) was used after the determination of the gel strength and viscosity. A gelatin solution was introduced into a glass capillary of 3.0 mm in diameter to form a column at a height of 6.0 ± 1.0 mm. The sample capillary was allowed to cool at 10.0 ± 0.1 °C for 17 h (sol–gel transition). The capillary was then placed in a water bath at 10.0 ± 0.5 °C so that it was completely immersed. The water bath was heated at 2 °C/min, and the gelatin column in the capillary was monitored. The temperature at which the gelatin column began to move in the capillary (gel–sol transition) was recorded as the melting point.

#### 2.5.8. Gelling Point

The Ninan method [40] with slight modifications was used to estimate the gelling point. A solution of gelatin was used at the same concentration (6.67%) as after the estimation of gel strength and viscosity. The gelatin solution in the test tube was placed in a water bath. After warming to 35.0 ± 0.5 °C, ice water was added to the water bath so that the cooling rate of the gelatin solution in the tube was 2 °C/min. Each time the temperature dropped by 0.5 °C, a 0.10 g metal ball was inserted into the tube. The temperature at which the ball got stuck in or on the gelatin solution layer was recorded as a gelling point.

#### 2.5.9. Digestibility

To determine digestibility, a procedure by Misurcova et al. [41] was used. First, the gelatins were ground into a fine powder. The sieve analysis revealed the particle size of ground gelatin, which ranged from 250 μm to 1.0 mm. Before measuring digestibility, it was necessary to estimate the dry matter and ash content of the sample (according to Equations (1) and (5)). Digestibility bags were washed in acetone and vented in a fume hood; 0.2500 g of gelatin was weighed in the bags and subsequently sealed with a table sealer. The prepared bags were placed in an incubation bottle and filled with 1.7 L of 0.1 mol/L HCl with 2.4 g of pepsin. The bottles were placed in a Daisy incubator (Dräger Medical, Lübeck, Germany) for 2 and 4 h. After time had elapsed, the bags were thoroughly washed with distilled water, and for some bags, the experiment ended and the remaining bags were put back in the incubation bottle, where they were covered with phosphate buffer of pH 7.45 and 2.40 g of pancreatin were added to the buffer. The bottle was incubated for 24 h at 37.0 ± 1.0 °C. The first incubation simulates the digestion process in the stomach for 2 or 4 h, and the second incubation simulates the digestion process in the intestinal tract for 24 h. After incubation, the bags were removed, thoroughly rinsed with distilled water and placed in a drying oven for 24 h at 105.0 ± 2.0 °C. After a day, the bags were removed, weighed, and placed in pre-annealed crucibles for ash determination. The digestibility of the dry matter of the sample (DMD) (%) was calculated according to Equations (7) and (8), where *DMR* is the weight of the sample without bag after incubation and drying (g); *DM* is the dry matter content of the sample (g); *c*_1_ is the correction of the weight of the bag after hydrolysis (g); *m*_1_ is the weight of the bag (g); *m*_2_ is the weight of the sample (g); *m*_3_ is the weight of the dried sample bag after incubation (g). Equations (9)–(11) refer to the calculation of the digestibility of organic matter (OMD) of the sample (%), where *AR* is the weight of the sample without the bag (g); *OM* is the content of organic matter in the dry matter of the sample (g); *c*_2_ is the correction of the bag after burning (g); *m*_1_ is the weight of the bag (g); *m*_2_ is the weight of the sample (g); *m*_4_ is the weight of the ash of the dry sample bag after incubation (g); *DM* is the dry matter content of the sample (%), and *AS* is the ash content of the sample (%).

## 3. Results

### 3.1. Yields of Gelatins and Hydrolysates

Table 2 shows a schedule of experiments with technological conditions and characterization of the process according to the Taguchi design with 2 factors at 3 levels and one blind experiment without enzyme addition. The table includes yields of hydrolysates, yields of the first, second, and third fractions of gelatin, and values of undissolved residue and calculated balance measurement errors. The amount of hydrolysate was in the 8.2 to 9.8% range. The lowest value, i.e., 8.2%, was obtained by experiment no. 8 and 9 and vice versa the highest value (9.8%) by experiment no. 1. For the blind experiment (no. 10), the amount of hydrolysate was only 1.7%. Processing the initial material with an enzyme seems to have the same effect on poultry tissue as the hydrolysate yield was negligible in experiment no. 10 (no enzyme was used here to process the initial material). The gelatin yield of the first fraction was between the values of 59.8 and 70.6%. A lower value was extracted for experiments 3, 6 and 9; in contrast, the highest value was found in the first experiment. Again, a low yield of 7.4% in experiment no. 10 confirmed that enzyme treatment has an effect on the yield of gelatin fractions. The average yield of the second fractions was 6.5 ± 1.3% and that of the third fraction was 1.4–3.8%. A blind experiment yielded only 5.4 and 0.1% for the second and third gelatin fractions. The amount of undissolved solid residue was in the range of 6.6 to 11.8%, and the value was used mainly to calculate the total balance according to Equation (3). Subsequently, BE was calculated according to Equation (4).

The regression equation for the gelatin yield of the first fraction was the following:(12)$$Yield\;of\;gelatin\;\left(\%\right)=141.0-311.3\;A-1.178\;B+4.53\;A\times B$$

According to the p-factor, the significance level for factor A was 0.009, for factor B 0.001, and for factors A and B 0.010. With 95% probability, we can say that both factor A and factor B, as well as the combination of factors A and B, are important for the extraction yield of the first gelatin fraction. Factor B (extraction temperature) had the greatest influence on the yield of the first gelatin fraction.

It can be seen in Figure 1 that the amount of gelatin extracted from the first fraction is affected by both the amount of enzyme added and the extraction temperature. It is true that the larger the amount of enzyme added or the higher the extraction temperature, the lower the yield of the first gelatin fraction. This phenomenon is apparently caused by a higher denaturation of collagen, i.e., a higher degree of bond breakdown. If we want to extract gelatin with the highest possible yield, it is desirable to use the smallest possible amount of enzyme and extract the gelatin at a lower extraction temperature. The graph shows that, when using 0.10% enzyme at a temperature of 55.0 °C, the extraction efficiency is more than 70.0%.

### 3.2. Gelatins Properties

Table 3 summarizes the schedule of experiments with technological conditions and the characterization of gelatin fractions. The necessary analyses were performed on all gelatins from the first fractions (ash content, gelatin gel strength, melting point, gelling point, clarity, and viscosity). Due to the lack of gelatin in the second and third fractions, all samples of the second and third fractions were mixed (each separately). The same analyses were performed on the gelatin mixtures of the second and third fractions as for the gelatins of the first fractions; the results are shown in the same table (exp. no. 11 and 12).

For applications in food, cosmetics, pharmacy, and medicine, the ash content of gelatins should be less than 2.0% [42]. As we can see, this condition was met for all samples. The ash content varied from 0.44 ± 0.02 (exp. no. 1) to 0.81 ± 0.02% (exp. no. 9), and the amount of ash was the lowest in the blind experiment (0.36 ± 0.02%). For the gelatin mixtures of the second and third fractions, the amount of ash was 0.937 ± 0.016 and 1.08 ± 0.02%, respectively. Despite the higher ash content values obtained, these are high-quality gelatins that can also be used in the pharmaceutical industry. Another important factor in the quality of gelatin is the strength of the gelatin gel. Some gelatins were found to have relatively low gel strength (25 ± 1 Bloom; exp. no. 7), and some had relatively high gel strength (439 ± 6 Bloom; exp. no. 6). On average, the gel strength of the first gelatin fractions was 241 ± 4 Bloom; the gel strength of the mixture of the second and third gelatin fractions was 172 ± 3 and 244 ± 5 Bloom, respectively. In the blind experiment (exp. no. 10), no gel strength was measured. The viscosity of the gelatin solution of the first fraction was in the interval from 1.0 ± 0.4 (exp. no. 7) to 3.4 ± 0.3 mPa·s (exp. no. 3 and 6). For the gelatin mixture of the second and third fractions, the viscosity was measured as 1.8 ± 0.2, resp. 2.16 ± 0.17 mPa·s. Other parameters investigated included the melting and gelling points. The gelling point of the first gelatin fractions was on average 19 ± 2 °C; the highest measured value was 22 ± 2 (exp. no. 3) and 22 ± 1 °C (exp. no. 6), and the lowest gelling point was found in exp. no. 7 (14 ± 2 °C). For the gelatin mixtures of the second fraction, the gelling point was 20 ± 1 °C and for the third fraction 17 ± 1 °C. The melting point of the first gelatin fractions was, on average, 32 ± 1 °C; the highest measured values were 37 ± 1 (exp. no. 3) and 37 ± 2 °C (exp. no. 6), and the lowest melting point was found in exp. no. 7 (28 ± 1 °C). The temperature of the gelatin mixtures of the second fraction was 32 ± 1 °C and that of the third fraction 33 ± 1 °C. Among the last analyses was the determination of the clarity of the gelatin gel, which for the gelatin of the first fractions was in the range from 0.88 ± 0.02 (exp. no. 3) to 2.059 ± 0.008 AU (exp. no. 7). This means that gelatin extracted under the conditions of 0.2% enzyme and 55.0 °C is the least clear or the cloudiest. For the gelatin mixture of the second fraction, the clarity was 1.946 ± 004 AU, and for the gelatin mixture of the third fraction, the clarity was 1.76 ± 0.09 AU.

#### 3.2.1. Gelatin Gel Strength and Viscosity

The regression equation for the gel strength of the first gelatin fraction was as follows:(13)$$Gel\;strength\;\left(Bloom\right)=-1242+750\;A+25.5\;B-24.0\;A\times B$$

According to the p-factor, the level of significance for factor A was 0.199, for factor B 0.001, and for factors A and B 0.733. With 95% probability, we can say that only factor B, i.e., the extraction temperature, is important for the gel strength of the gelatin of the first fraction. Factor A and the combination of factors A and B do not have a significant effect on gel strength.

The regression equation of the viscosity of the first fraction was as follows:(14)$$Viscosity\;\left(\mathrm{mPa}\cdot s\right)=-7.31+20.3\;A+0.1692\;B-0.447\;A\times B$$

According to the p-factor, the significance level for factor A was 0.048, for factor B 0.003, and for factors A and B 0.390. With 95% probability, we can say that the viscosity of the first gelatin fraction is mainly influenced by factors B and A. The combination of factors A and B does not have an effect on viscosity.

Figure 2a shows a contour graph of the effects of factors A and B on the strength of the gelatin gel. Herein, the growth of the gel strength in Blooms is evident with increasing extraction temperature; at 70.0 °C, a gelatin with a gel strength of more than 400 Bloom was extracted. The effect of enzymes on gel strength tends to increase and decrease. The best strength characteristics of the gel were achieved at 0.15% enzyme. If we start to add or reduce the amount of enzyme, the strength of the gel will decrease. This phenomenon may be caused by the shortening of the collagen chain length of gelatin, and thus the reduction of the collagen molecular weight and the shortening of the amino acid chain length, which results in a decrease in gel strength. At the same time, a higher gel strength is related to a higher proportion of α and β chain components. A higher gel strength can also be caused by the presence of hydroxyproline in collagen, which causes a better stability of the hydrogen bonds between water molecules and the free hydroxyl groups of amino acids in gelatin. The best gel strength, 439 ± 6 Bloom, was measured at the highest extraction temperature (70.0 °C) and the addition of medium enzyme (0.15%).

Figure 2b shows a contour graph of the effects of factors A and B on viscosity; the viscosity decreased as the amount of enzyme increased, but viscosity increased as the extraction temperature increased. Viscosity is directly proportional to the strength of the gel, that is, the higher the gel strength, the higher the viscosity, which is influenced by the molecular weight, the length of the amino acid chain, and the higher proportion of α and β chain components. Low viscosity gives brittle gels, whereas high viscosity gelatin gives harder, more extensible gels, and it is because of this that low viscosity is also associated with excessive and unwanted collagen hydrolysis. The highest viscosity was measured at exp. no. 3 and 6 (the values were the same, 3.4 ± 0.3 mPa·s), and in both cases the gel strength was more than 400 Bloom.

#### 3.2.2. Melting and Gelling Points of Gelatins

The regression equation for the melting point of the first fraction was as follows:(15)$$Melting\;point\;\left({}^{\circ}C\right)=-12.1+101.7\;A+0.767\;B-2.00\;A\times B$$

According to the p-factor, the significance level for factor A was 0.049, for factor B 0.001, and for factors A and B 0.232. With 95% probability, we can say that the melting point of the first gelatin fraction is mainly influenced by factor B and factor A. The influence of the combination of factors A and B on the melting point of gelatin has no effect.

The regression equation for the gelling point of the first gelatin fraction was as follows:(16)$$Gelling\;point\;\left({}^{\circ}C\right)=-9.0-0\;A+0.444\;B+0\;A\times B$$

According to the p-factor, the significance level for factor A was 1.000, for factor B 0.003, and for factors A and B 1.000. With 95% probability, we can say that only factor B, i.e., the extraction temperature, is important for the gelling point of the first gelatin fraction. The influence of factor A and the combination of factors A and B has no effect on the gelling point.

In Figure 3a, we can observe the influence of the contour effects of factors A and B on the melting point of the gelatin gel. Herein, with a lower amount of enzyme added and at a lower extraction temperature, the gelatin fraction with a lower melting point of the gel is extracted. This is related to the strength of the gel, because the higher the strength of the gel, the higher the melting point and viscosity. Therefore, in experiments no. 3 and 6, wherein the highest gel strength was measured (409 ± 6 and 439 ± 6 Bloom), the melting point was also the highest. Values were 37 ± 1 °C for exp. no. 3 and 37 ± 2 °C for exp. no. 6. On the contrary, the worst gel strength was found in exp. no. 7 (25 ± 1 Bloom), and at the same time, the lowest melting point was measured in this experiment, 28 ± 1 °C.

In Figure 3b, we can observe the influence of the contour effects of factors A and B on the gelling point of the gelatin gel. As the extraction temperature increased, the stiffness of the gelatin gel also increased, and again the gelling point depends on the gel strength. It is true that the lower the gel strength, the lower the gelling point. At the lowest gel strength, 25 ± 1 Bloom (exp. no. 7), the gelling point was 14 ± 2 °C, and on the contrary, at the highest gel strength, 409 ± 6 and 439 ± 6 Bloom (experiments no. 3 and 6), it was 22 ± 2 and 22 ± 1 °C, respectively.

#### 3.2.3. Ash Content and Clarity of Gelatins

The regression equation for the ash content of the first gelatin fraction was as follows:(17)$$Ash\;content\;\left(\%\right)=-0.448+4.37\;A+0.0095\;B-0.0193\;A\times B$$

According to the p-factor, the significance level for factor A was 0.001, for factor B 0.067, and for factors A and B 0.792. With 95% probability, we can say that factor A, the amount of enzyme added, has a significant effect on the ash content of the first gelatin fraction. The influence of factor B and the combination of factors A and B has no effect on the ash content.

The regression equation for the clarity of the first fraction was as follows:(18)$$Clarity\;\left(A\right)=1.40+19.5\;A-0.0115\;B-0.244\;A\times B$$

According to the p-factor, the significance level for factor A was 0.010, for factor B 0.001, and for factors A and B 0.215. With 95% probability, we can say that both factor A and factor B are important for the clarity of the gelatin solution of the first fraction. The effect of the combination of factors A and B on clarity is not essential.

In Figure 4a, we can see a contour graph of the effects of factors A and B on the ash content. It can be seen from the figure that, as the amount of added enzyme increases and the extraction temperature increases, so does the amount of ash content in the gelatins. The highest value of the ash content was found with 0.20% enzyme and extraction temperature of 70.0 °C, and the value was 0.81 ± 0.02%.

In Figure 4b, we can see a contour graph of the effects of factors A and B on the clarity of the gelatin solution. Herein, as the amount of enzyme decreased and the extraction temperature increased, the gelatin solution had better clarity. The least turbid was gelatin in exp. no. 3 (0.10% enzyme and 70.0 °C extraction temperature), and the clarity was 0.88 ± 0.02 AU. The turbidity of gelatin depends on the nature of the initial raw materials as well as the chemicals used in its preparation. Clarity is affected by inorganic, protein, and mucosubstitution contaminants that enter the gelatin solution during preparation and especially during the extraction itself. However, the quality of the solution can be improved by filtering through filter paper, which traps unwanted contaminants. Therefore, gelatin in exp. no. 7 (0.20% enzyme and extraction temperature of 55.0 °C) has such poor clarity, 2.059 ± 0.008 AU, because it contains undesirable contaminants.

### 3.3. Digestibility of Gelatins

Table 4 shows the digestibility results of the gelatins of the first fractions, including the digestibility of the hydrolysate and the gelatin mixtures of the second and third fractions. First, the processing was carried out in pepsin, which simulates the digestion process in the stomach. Processing was carried out for 2 h and 4 h. Since some gelatins were not fully digested, another analysis was performed, processing both in pepsin (simulation of the digestion process in the stomach for 4 h) and subsequently in pancreatin, which simulates the digestion process in the intestinal tract for 24 h. The DMD values (%) represent the dry matter values of the sample without the bag after incubation and drying; OMD values (%) represent the digestibility of the organic matter.

When processing the samples in pepsin for 2 h, the gelatins of experiments no. 4, 8 and 9 (and the hydrolysate, exp. no. 10) had 100.00% digestion of the sample according to the DMD values or according to the dry matter values of the sample. However, according to the OMD values, i.e., the digestibility of organic matter, 100.00% digestion occurred even with gelatin experiments no. 1 and 2. Since this first stage did not cause 100.00% digestion for all gelatins, a simulation was performed with double the processing time, that is, for 4 h. Herein are 100.00% digestibility values according to DMD for all gelatins and hydrolysates, except gelatin exp. no. 7. Even the gelatin mixture of the second fraction represents 100.00% digestion, and for the gelatin mixture of the third fraction, the dry matter values of the sample were 99.82%. In this case, the OMD values are the same as the DMD values. Since complete digestion did not occur even after 4 h of the process, the simulation was carried out in both the stomach (for 4 h) and the intestine (for 24 h). A single gelatin exp. no. 7 was not 100.00% digested according to the DMD values, but when converted to OMD organic matter, 100.00% digestion occurred with this gelatin as well. This is apparently due to the fact that gelatin has a relatively high molecular weight. In general, it can be said that gelatins extracted from poultry stomachs after previous processing in a proteolytic enzyme are of high quality and not burdensome for the organism, as their digestibility is 100.00%.

## 4. Discussion

### 4.1. Yields of Gelatin

The yield of gelatin is generally dependent on the type and age of the tissue and also on the degree of intermolecular crosslinking and the type of collagen [19]. During the 12–14 h processing of broiler bones in H_3_PO_4_ with a concentration of 8–10%, it was found that the gelatin yield was only 8.53–14.60%, depending on the age of the broiler [21]. In our study, the gelatin yield of the first fraction was between 59.8 and 70.6%. This might be caused by a lower level of intermolecular crosslinking in the chicken stomach collagen in comparison to the broiler bone collagen and also by a different process of conditioning and gelatin extraction. A study [3] focused on the extraction of gelatin from the skin of stingrays using two acids (HCl and CH_3_COOH, concentration 0.01–0.20 mol/L) and found that the yield is very low compared to the extraction of gelatin from poultry stomachs. For HCl, it was only 2.8–5.5%, and, for CH_3_COOH, it was slightly higher (4.1–7.0%). In another study, in which they processed poultry paws using 0.1 mol/L of HCl, they found that, with increasing extraction temperature, the efficiency also increases from 75.0% (extraction at 65 °C) to 90.0% (extraction at 95 °C). This might be due to a higher degree of hydrolysis at higher extraction temperatures [22]. We managed to extract a maximum of 70.6% gelatin (processing in 0.10% enzyme and extraction temperature of 55.0 °C), which is a very good yield from an economic point of view of the process [5,19]. Enzymatic conditioning of poultry tissue (feet, heads, skin) results in a higher degree of the conversion of collagen into gelatins (on average 58 ± 12%) [23,34,43] compared to the conditioning of collagen tissue in acids or alkalis [9]. Additionally, the yield of gelatin generally depends on the age of the tissue. Under similar processing conditions, bovine and pork tissues give lower gelatin yields compared to poultry or fish tissues [16,20].

### 4.2. Gelatin Properties

Gelatin ash content. Very low-quality gelatins with an ash content of up to 31.5% were extracted from broiler chicken bones after processing in H_3_PO_4_ [21]. The ash content was more than ten times higher than the maximum limit allowed for the application of gelatin in food, pharmacy, or cosmetics [42,44]. In our study, high-quality gelatins with a very low ash content (0.72 ± 0.02%) and high gel strength (439 ± 6 Bloom) were prepared.

Gelatin gel strength. In a study [4], gelatins from mechanically deboned chicken meat residues were extracted using HCl with gel strengths ranging from 320 to 570 Bloom. However, the study did not indicate the content of ash in the gelatins. If the ash content is less than 2.0% within the study, high-quality gelatin can be applied to the production of hard gelatin capsules or to the production of collagen films that can be applied to burns [17]. The 182–360 Bloom gelatins were extracted from HCl conditioned chicken paws collagen [22]. Gelatins with very high gel strength were extracted from the stingray skin using HCl (620 Bloom) and CH_3_COOH (650 Bloom) [3]. Differences in gel strength values may be associated with the different amino acid profile of each collagen tissue, as well as the different distribution of oligomers and fragments with low molecular weight in gelatin [9,21]. Gelatin prepared from NaOH-treated tuna skin showed 207 Bloom gel strength [20]. Different amino acid compositions, especially lower hydroxyproline content, lower the gel strength of gelatin. This is due to insufficient hydrogen bonding between the water molecules and the free carboxyl groups in the gelatin.

Viscosity of gelatins. Gelatins prepared from mechanically deboned chicken meat residues have a higher viscosity (2.82 to 5.80 mPa·s) [4] compared to our gelatins (1.0 to 3.4 mPa·s). The viscosity values are directly proportional to the strength of the gel; higher gel strength results in higher viscosity. For gelatins prepared from chicken paws collagen conditioned in HCl, high viscosity values (5.12–7.61 mPa·s) [22] were reported. Lower viscosity values might be caused by collagen conditioning, during which the cleavage of peptide bonds in the primary structure of weakly cross-linked chicken collagen occurs. To conclude, viscosity can be influenced not only by the type of initial tissue but also by the type of collagen conditioning and by the conditions of the gelatin extraction process (pH, temperature, time) [9,44].

Gelatins melting point. When extracting collagen from chicken paws conditioned with HCl, the melting point values were 32.1–37.5 °C [22]. Gelatin extracted from tuna skin using NaOH showed melting points 31–35 °C [20]. The highest melting point in our study was 37 ± 2 °C. It is evident that the melting point is related to the strength of the gelatin gel.

### 4.3. Digestibility of Gelatins

Generally, the digestibility of gelatins, regardless of the type of gelatin, the chosen conditioning, and extraction method, is more than 95.0% [45,46]. However, some studies reported that the digestibility of gelatin is related to its molecular weight. This is the case for gelatins prepared from salmon skins and fish trimmings, wherein the digestibility was only 85.9%, which can be attributed to the higher molecular weight of the gelatins [47]. Compared to these studies, our study demonstrates the 100% digestibility of gelatins. We assume that, in addition to the molecular weight of gelatins, the enzyme conditioning of the raw material might also influence the complete digestibility.

### 4.4. Proposed Applications of Chicken Stomach Gelatin

In the food industry, low gel strength (<100 Bloom) can be used for the production of confectionery such as caramels, licorice, deposited marshmallows, and meringues. Gelatins with medium gel strength (100–250 Bloom) are suitable for the production of gummy bears, aspics, jellies, dairy products, extruded marshmallows, and desserts. These two gelatin groups exhibit melting points between 28 and 33 °C. Furthermore, gelatins from both groups are valued in the production of nutritional supplements [16,44]. Gelatins with high gel strength (250–400 Bloom) could possibly be used in the production of soft and hard pharmaceutical gelatin capsules. These melting points of gelatins are approximately 34 °C. Gelatins with a very higher Bloom value (>400) can be applied in the production of collagen fibers or films for medical purposes or for the production of contact lenses [9,44]; the melting point of these gelatins reaches up to 37 °C.

### 4.5. Evaluation of the Importance of Work for Science and Practice

Gelatin of types A and B is produced in industrial practice. The presented study deals with the extraction of gelatin from poultry stomachs after previous collagen enzyme conditioning. A microbial endopeptidase Protamex^®^ provides the partial denaturation of the quaternary structure of collagen. Under suitable conditions during the conditioning process (time, temperature, amount of enzyme) covalent lysine–aldehyde-based bonds (Schiff bases) between individual tropocollagen molecules are cleaved. During processing at room temperature (22.0 ± 1.5 °C), the unwanted cleavage of peptide bonds is avoided. Before the hot-water extraction of gelatin (thermal denaturation of collagen), the enzyme is removed by washing the conditioned collagen with water. This processing step minimizes the hydrolysis of peptide bonds in individual chains of tropocollagen molecules during gelatin extraction at higher temperatures, which would result in lowering the molecular weight of the peptide chains of the prepared gelatin, thus affecting the gelling properties and viscosity of the gelatin. This innovative approach turns out to be the most suitable method of gelatin production in terms of saving time and energy; this method is also environmentally friendly [23].

The gelatin yield of the first fraction was on average 63 ± 4% within the study. Gel strength ranged from 25 to 439 Bloom. The ash content was 0.44 to 0.81%, and the melting point of the gelatin gel was on average 32 ± 3 °C. Taking into account the yield of gelatin, gel strength, melting point, and ash content, the use of 0.10% enzyme at an extraction temperature of 62.5 °C were optimal conditions for the preparation of gelatin from chicken stomach collagen. The yield of the obtained gelatin is 62.4%, a gel strength of 275 Bloom, a gelling point of 33 °C, and an ash content of 0.49%. Gelatin would find application, e.g., in the production of jelly, marshmallows, aspics, and dairy products. The optimal conditions to prepare gelatin suitable for the production of hard gelatin capsules or collagen films [16,44] are the 0.10% enzyme and the extraction temperature of 70.0 °C; the yield of the first fraction of gelatin is 59.8%, the gel strength 409 Bloom, the gelling point 37 °C, and the ash content 0.51%.

Approximately 130 million tons of chicken are slaughtered annually, of which the stomachs make up about 3.9 million tons [48]. Based on the assumption that one half of the world (especially in Asia) consumes stomachs and some portion do not pass the hygiene control in order to be consumed, approx. 1.5 million tons of stomachs represent by-products suitable for production of gelatins. For applications in food and pharmacy, gelatins prepared according to our procedure do not need any purification step, as their ash content is <0.81%. For some specific applications of gelatin (e.g., photographic emulsions) further purification procedures (deionization, clarification) may be required [44,49].

## 5. Conclusions

The study deals with the biotechnological processing of chicken stomachs into gelatins. Initial tissue was treated with a microbial proteinase Protamex^®^. The main objective of the work was to verify the process conditions for gelatin preparation and to monitor the effect of selected process parameters on gelatin yields. The amount of enzyme added and the extraction temperature were monitored to estimate the overall efficiency of the process and the quality of the extracted gelatins. Gelatin model samples were prepared, and the strength of gelatin gels, ash content, viscosity, melting point, gelling point, and digestibility of gelatins were estimated. Optimal conditions for the processing of poultry stomachs into gelatins were proposed with respect to their application in the food and pharmaceutical industry. When appropriate technological conditions are chosen, it is possible to extract high-quality gelatins from chicken stomach collagen, which are fully comparable to standard gelatins from pork and beef. The work demonstrated a high potential to fulfill the philosophy of circular economy. With an optimized combination of technological conditions, it is possible to obtain high-quality gelatins (and hydrolysates) from chicken stomachs, which until now were just secondary animal by-products from poultry farms. Furthermore, the preparation process is economically beneficial and environmentally friendly and could replace the traditional acid and alkaline method of processing collagen tissues into gelatins in the future.

## Data Availability

The data are available from the corresponding author.

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
