# Peer review of "Preparation of Gelatin from Broiler Chicken Stomach Collagen"

_foods, 2022, doi:10.3390/foods12010127_

Round 1
Reviewer 1 Report
1. Sentence rephrasing is required throughout the manuscript. Every sentence requires grammatical corrections and rephrasing of sentences.
2. Introduction is very lengthy and unnecessary. It would be better if the authors make the introduction very specific. Some of the details can be moved discussion part.
3. Introduction has focussed too much on financial aspects of gelatin extraction from chicken, but that is already happening using other parts of chicken such as skin, bone etc which are better source of gelatin. It will be apt for the authors to simply modify the introduction significantly.
4. The hypothesis and objectives section has to be brief and to the point. It needs significant modification.
5. The processing of stomach for preparation of gelatin extraction itself includes all kinds of alkali and organic solvent treatment. So, how significant difference will it make if only the final extraction is done by enzymatic treatment?
6. It would be more appropriate if the authors can add a flow chart of the extraction process and enzymatic treatment.
7. There is lack of clarity in subsection 2.3. Processing of Chicken Stomachs into Gelatins
8. Experimental design should come before the actual experiment methodology.
9. There is lack of flow in the arrangement of equation formulae for example, extraction efficiency (equation 1) requires information of DMS but DMS calculation is shown in equation 5.
10. The authors have used “determined” term in methodology, however, at various places that term would be scientifically wrong, the right term would be “estimated”.
11. A step for confirmation of gelatin extraction and purification may add value to the paper.
12. Since the author has discussed that the yield of gelatin is generally dependent on the type and age of the tissue, have they taken into consideration the age of chicken stomach taken before starting the experiment?
13. Due to lack of scientific language clarity, the discussion section is not having the readability flow.
Author Response
Dear Sir / Madam,
Thank you very much for your comments and recommendations referring to our manuscripts ant its improvement. We did our best to revise our paper according to your suggestions as well as suggestions of another reviewer. Changes in the manuscript are made in red color. Point-by-point responses to your comments are below.
Yours faithfully,
Pavel Mokrejš (corresponding author)
Reviewer Comment No.1: Sentence rephrasing is required throughout the manuscript. Every sentence requires grammatical corrections and rephrasing of sentences.
Response: Many sentences were rephrased. The whole manuscript was checked with the help of a colleague with English advanced level. Then, the text was checked and corrected again with the help of Writefull (a tool for advanced language correction of English texts).
Reviewer Comment No. 2: Introduction is very lengthy and unnecessary. It would be better if the authors make the introduction very specific. Some of the details can be moved discussion part.
Response: Introduction was shortened. Details referring to chicken stomachs have been moved to “Discussion” part (new references were added). Please see last but one paragraph in chapter 4.5.
Reviewer Comment No. 3: Introduction has focussed too much on financial aspects of gelatin extraction from chicken, but that is already happening using other parts of chicken such as skin, bone etc. which are better source of gelatin. It will be apt for the authors to simply modify the introduction significantly.
Response: Financial aspects of gelatin extraction from chicken were modified. Introduction newly presents important initials referring to the topic. We agree that chicken skins and bones are more common source of collagen for gelatin preparation. More literature about extraction conditions and gelatin properties from these tissues is available. That is not true for chicken stomachs. That was the point why our research group focused on processing of (purified) collagen from chicken stomachs. Moreover, new method (previously patented by our team for processing of chicken feet into gelatins), based on enzyme conditioning of collagen, to induce denaturation of quaternary structure of collagen, was applied.
Reviewer Comment No. 4: The hypothesis and objectives section has to be brief and to the point. It needs significant modification.
Response: Objectives and hypothesis of the study are modified and shortened.
Reviewer Comment No. 5: The processing of stomach for preparation of gelatin extraction itself includes all kinds of alkali and organic solvent treatment. So, how significant difference will it make if only the final extraction is done by enzymatic treatment?
Response: Yes, this purification procedure of starting raw material (chicken stomachs) is necessary; for details please see first paragraph in chapter 2.4. With water, weak solutions of NaCl, NaOH non-collagenous parts from the raw material are washed out. Then, fat needs to be removed (with solvents). This purification procedure is required for any kind of processed collagenous animal tissues prior to its processing into collagen products (gelatins, hydrolysates, peptides). The purification procedure was used from literature and adopted to chicken stomachs based on our previous experience with processing chicken skins, paws, heads etc. to minimize the amount of water, amount and concentration of chemicals, and processing time. Without these purification steps, the yield of gelatins will be lower, especially due to the presence of fat. Moreover, gelatin quality will be very low. All the “impurities” and some portion of fat will be present in gelatin solution. Thus, afterwards, gelatin solution would have gone through complicated purification procedures to separate these unwanted parts.
Reviewer Comment No. 6: It would be more appropriate if the authors can add a flow chart of the extraction process and enzymatic treatment.
Response: Flow chart of processing chicken stomach into 3 gelatin fractions was added in chapter 2.4. „Processing of Chicken Stomachs into Gelatins“.
Reviewer Comment No. 7: There is lack of clarity in subsection 2.3. Processing of Chicken Stomachs into Gelatins
Response: With respect to previous comments, we hope that addition of flow chart showing key steps of processing chicken stomach into gelatin fractions will clarify the process for the readers. Processing of chicken stomachs into gelatins was re-written and simplified. Remark: The methodology was modified as suggested by another reviewer - each method for gelatin testing is presented with a reference in separate subheading.
Reviewer Comment No. 8: Experimental design should come before the actual experiment methodology.
Response: Experimental design (chapter 2.3.) is newly placed before the actual experiment methodology (chapter 2.4.). Moreover, each method for gelatin testing is presented with in separate subheading (as suggested by another reviewer).
Reviewer Comment No. 9: There is lack of flow in the arrangement of equation formulae for example, extraction efficiency (equation 1) requires information of DMS but DMS calculation is shown in equation 5.
Response: Equations were checked, clarified and re-arranged.
Reviewer Comment No. 10: The authors have used “determined” term in methodology, however, at various places that term would be scientifically wrong, the right term would be “estimated”.
Response: Thank you for your comment. In chapter 2.3. “Experimental Design and Calculation of the Results“ term “determined” is replaced with term “estimated” where appropriate.
Reviewer Comment No. 11: A step for confirmation of gelatin extraction and purification may add value to the paper.
Response: Confirmation of (total) gelatin extraction yield is mentioned in last but one (newly added) paragraph in chapter “4.5. Evaluation of the Importance of Work for Science and Practice“. Regarding the purification of gelatins. All gelatins prepared from chicken stomach collagen contain very low amount of minerals - less then 0.8 %, which is suitable for all pharmaceutical and food application. Maximum permissible limit, according to standards, is 2.0 % (reference [42]). Nevertheless, some facts regarding further purification steps are mentioned in the same paragraph.
Reviewer Comment No. 12: Since the author has discussed that the yield of gelatin is generally dependent on the type and age of the tissue, have they taken into consideration the age of chicken stomach taken before starting the experiment?
Response: Yes, this fact has been taken into consideration. Information regarding the breed type and the age of chicken was implemented in chapter 2.2. (Ross 708 broiler chicken stomachs, aged 35 days). In this respect, the gelatin yield versus the age of initial tissue is mentioned in chapter 4.1.
Reviewer Comment No. 13: Due to lack of scientific language clarity, the discussion section is not having the readability flow.
Response: Sentences in discussion sections were modified. Two new references were added.
Reviewer 2 Report
The manuscript titled “Chicken stomachs as a new source of collagen for preparation of gelatins” described processing as well as gelatin development processes using chicken stomach, which could help food industries as well as pharma industry people to develop more useful product. The article is been written well, however it needs minor revision in further submission. Reviewer comments has been marked in pdf file, kindly refer to pdf pages.

Author Response
Dear Sir / Madam,
Thank you very much for your comments and recommendations referring to our manuscripts ant its improvement. We did our best to revise our paper according to your suggestions as well as suggestions of another reviewer. Changes in the manuscript are made in red color. Point-by-point responses to your comments are included in pdf file and below as well.
Yours faithfully,
Pavel Mokrejš (corresponding author)
Reviewer Comment to the title: Author should reframe the title of the manuscript. Since at first view it gives an impression of review article.
Response: The title was changed so that more emphasis is given on experimental point of view.
Reviewer Comments referring to Abstract: Authors are suggested to write the method used for this experiment after the objective of the abstract. Authors are suggested to write objective with better clarity.... Kindly check line no 16. Simply write what characteristic going to measure delete above all ...? Some typographical comments.
Response: Some parts of the Abstract were modified. Firstly, the objectives are presented. Secondly, it is cleared what gelatins characteristics were measured. Then, description of experimental method is given. Moreover, some typographical errors were corrected.
Reviewer Comment regarding to Abstract: “Bloom”
Response: The gel strength of gelatin measures the rigidity of a gel formed from a 6.67% solution prepared according to prescribed conditions. The Bloom value is measured as a force (weight in grams) required to depress prescribed area of the sample into the depth of 4 mm using texture analyzer. The details of procedure are described in: Standard testing methods for edible gelatin - Official Procedure of the Gelatin Manufacturers Institute of America, Inc.; please see reference [38].
Reviewer Comment referring to Introduction: Introduction needs to be shortened and important initial should be only presented with better clarity of the objective.
Response: Introduction was revised. It is shorter and more specific. Objectives and hypothesis of the study are modified and to the point of (as suggested by another reviewer).
Reviewer Comments to “2. Materials and Methods”: Each methods used for testing must be written in separate subheading. It is going to be very difficult for the reader to understand the methodology used. Each method used should be presented with reference.
Response: The methodology was modified as suggested – each method for gelatin testing is presented with a reference in separate subheading; 3 new references referring to testing methods of gelatins were added. Moreover, experimental design (chapter 2.3.) is newly placed before the actual experiment methodology (chapter 2.4.) - as suggested by another reviewer. Further, flow chart showing extraction process of gelatins from chicken stomach collagen is added (as suggested by another reviewer).
Round 2
Reviewer 1 Report
1. There are still many scientific errors that needs correction. One such instance is in line 33 “Collagen is the basic building element of all types of connective tissues”. Collagen is not a building element, rather a protein biomolecule. Likewise in line 39, “ Collagen can be hydrolyzed more gently by enzymes”, the word gently is not apt for the sentence. Overall scientific language is missing in the entire manuscript.
2. There has not been up to mark in language improvisation, hence, readability is being affected in order to understand the message of the sentences.
3. Rephrasing is required in line 77 “The objective of the work was to verify the process conditions”, it should be “The objective of the study IS to verify….”. Essentially, the objectives should be in present tense.
4. The authors need not to explain “what is Design of Experiments” as mentioned in line 124, “Design of experiments (DOE) is a mathematical tool that quantifies….”. They can straight away explain their design of experiments in order to meet their objectives. Entire first paragraph may be removed.
5. Table 1 requires further reshuffling as part of sequential outcomes, beginning with DM, for example.
6. Experimental design and Calculation of results should be kept in separate subsections.
7. Subsection 2.4. Processing of Chicken Stomachs into Gelatins should come before experimental design. Calculation of results can follow experimental design.
8. How does protease enzymes degrade collagen but not gelatin as both are proteins. Give sufficient justifications.
Author Response
Dear Sir / Madam,
Thank you for your comments and recommendations. We revised our paper a second time according to your suggestions. New changes in the manuscript (after Round 2) are made in green color; changes in the manuscript after Round 1 are made in red. Some responses to your comments are included below.
Reviewer Comment No.1: There are still many scientific errors that needs correction. One such instance is in line 33 “Collagen is the basic building element of all types of connective tissues”. Collagen is not a building element, rather a protein biomolecule. Likewise in line 39, “ Collagen can be hydrolyzed more gently by enzymes”, the word gently is not apt for the sentence. Overall scientific language is missing in the entire manuscript.
Response: The first paragraph in the Introduction (collagen, enzymes) is rewritten.
Reviewer Comment No.2: There has not been up to mark in language improvisation, hence, readability is being affected in order to understand the message of the sentences.
Response: The manuscript was checked again with Writefull (academic English texts correction software). Some minor grammar corrections were made.
Reviewer Comment No. 3. Rephrasing is required in line 77 “The objective of the work was to verify the process conditions”, it should be “The objective of the study IS to verify….”. Essentially, the objectives should be in present tense.
Response: The objectives, as well as hypothesis, of the study are now in the present tense. We apologize for that. Notice: In the latest FOODS published articles both present and past tenses are used for expressing the objectives of the study; we had been wondering which tense to use…
Reviewer Comment No. 4. The authors need not to explain “what is Design of Experiments” as mentioned in line 124, “Design of experiments (DOE) is a mathematical tool that quantifies….”. They can straight away explain their design of experiments in order to meet their objectives. Entire first paragraph may be removed.
Response: The first paragraph that describes DOE was removed. Only the paragraph describing the design used during our experiments (Taguchi design) is included.
Reviewer Comment No. 5. Table 1 requires further reshuffling as part of sequential outcomes, beginning with DM, for example.
Response: Changes were made as suggested. The equations start with the dry matter calculation equation and followed with the yield of gelatins. A similar reshuffling was made for the balance error equation.
Reviewer Comment No.6. Experimental design and Calculation of results should be kept in separate subsections.
Response: Changes were made as suggested. After the revision, there are following chapters: „2.4. Experimental Design“; and finally, „2.5. Calculation of the Results“.
Reviewer Comment No. 7. Subsection 2.4. Processing of Chicken Stomachs into Gelatins should come before experimental design. Calculation of results can follow experimental design.
Response: Changes were made as suggested. Firstly, chapter “2.3. Processing of Chicken Stomachs into Gelatins“; then, „2.4. Experimental Design“; and finally, „2.5. Calculation of the Results“.
Reviewer Comment No. 8. How does protease enzymes degrade collagen but not gelatin as both are proteins. Give sufficient justifications.
Response: Information on the details of the Protamex® endoprotease used to conditioning collagen is given in last paragraphs of chapter 2.1. Furthermore, the mechanism of chemical denaturation (partial hydrolysis) of collagen with Protamex® is explained in the first paragraph of chapter 4.5.